# Effect of 2,5-Dicarbonyl-3-Isobutyl-Piperazine on 3-Isobutyl-2-Methoxypyrazine Biosynthesis in Wine Grape

**DOI:** 10.3390/foods12173258

**Published:** 2023-08-30

**Authors:** Yujuan Lei, Zhansheng Ma, Ping Wang, Xuchen Qin, Xueqiang Guan, Zhenwen Zhang

**Affiliations:** 1College of Food Science & Technology, Hebei Normal University of Science and Technology, Qinhuangdao 066004, China; leiyujuan@nwafu.edu.cn (Y.L.); mzs2020@sina.com (Z.M.);; 2College of Food Engineering, Shihezi University, Shihezi 832000, China; wangping92@sina.com; 3Shandong Academy of Grape/Shandong Technology Innovation Center of Wine Grape and Wine, Jinan 250100, China; 4College of Enology, Northwest A&F University, Yangling 712100, China

**Keywords:** 2,5-Dicarbonyl-3-isobutyl-piperazine, incorporation, grape, in *situ*, IBMP biosynthesis

## Abstract

The metabolic pathway of 3-alkyl-2-methoxypyrazines (MPs) in grape remains largely unclear except for the final step. In this study, the 2,5-dicarbonyl-3-isobutyl-piperazine (DCIP), which is proposed as the key intermediate of 3-isobutyl-2-methoxypyrazine (IBMP) biosynthesis, was incorporated into Cabernet Sauvignon clusters in *situ* using a soaking method. The IBMP concentration of grape and the expression patterns of VvOMTs in berry skin were monitored over two consecutive years. The results showed that the IBMP concentration of grape treated with DCIP was significantly increased at maturity in both years. The relative expression levels of *VvOMT1* and *VvOMT3* in berry skin were positively correlated with the IBMP accumulation. After DCIP incorporation, the relative expression level of *VvOMT1* and particularly that of *VvOMT3* were obviously up-regulated and closely mirrored the IBMP accumulation pattern in two consecutive years. Therefore, we speculate that DCIP may be a key intermediate involved in the biosynthesis of IBMP and plays an important role in regulating IBMP accumulation.

## 1. Introduction

3-Alkyl-2-methoxypyrazines (MPs) are kinds of nitrogen-containing heterocyclic compounds, which are considered to be one of the main sources of herbaceous and vegetal sensory characteristics of grapes and wine [1,2,3]. At present, seven MPs have been identified in grapes and wines [4], and 3-isobutyl-2-methoxypyrazine (IBMP), which imparts a green pepper aroma, is considered to be the most important one, due to the fact that its concentration is always significantly higher than the sensory threshold [5,6,7,8]. In fact, the concentrations of MPs in grapes and wines are typically low (2–30 ng·L^−1^) [9,10]. The reason why such low concentrations still have an important influence on wine aroma and flavor [11,12] is their sensory thresholds are also extremely low. Previous studies have shown that the sensory threshold of IBMP was 2 ng·L^−1^ in water, 1–6 ng·L^−1^ in white wine, and 10–16 ng·L^−1^ in red wine [6,7,8].

In view of the extremely low concentration of MPs in grapes and wines, one of the key points in this research field is the detection and quantitative methods developing. In recent years, many studies were involved in this field and great achievements were made, and many accurate and efficient techniques with higher sensitivity were developed [13]. Among them, isolation methods such as liquid–liquid extraction (LLE), solid-phase microextraction (SPME), headspace solid-phase microextraction (HS–SPME), and so forth, and clean-up procedures such as solid-phase extraction (SPE) provide powerful tools to detect MPs in concentrations on the ng·L^−1^ level [14,15]. And the quantitative analyses of MPs have used gas chromatography–mass spectroscopy (GC-MS), GC combined with time-of-flight mass spectrometry (TOF-MS), or GC combined with nitrogen phosphorous detectors (NPDs) and so on [13,16,17,18], which provide a methodological guarantee for different researchers to carry out relevant studies.

Generally speaking, excessive levels of MPs are unacceptable for both the wine maker and consumer due to the fact that it will make the wine be regarded as poor quality, which is made from the unripe grape [19,20]. The MPs’ concentration in wine mainly depends on the final concentration of MPs in the grape berries it is made from, because these compounds mainly exist in the grape berry and show almost no change during fermentation [21]. Therefore, the key to controlling the concentration of MPs in wine is to reduce their content in the grape berries.

Unfortunately, the biosynthesis pathway of MPs remains largely unclear [4]. To date, there are two hypotheses for MPs’ biosynthesis. One hypothesis proposed by Murray et al. [22] suggests that the initial step of MPs’ biosynthesis involves amidation of an amino acid, followed by condensation with glyoxal or glyoxylic acid to form 3-alkyl-2-hydroxypyrazines (HPs), and finally the HPs are methylated to MPs (hereinafter referred to as “Amino acid amidation pathway”). According to the “Amino acid amidation pathway”, the IBMP biosynthesis begins in the amidation of L-Leucine (L-Leu), that is, the L-Leu amides to 2-amino-4-methylpentanamide (AMPA), and the AMPA condensates with the glyoxal or glyoxylic acid to form 3-isobutyl-2-hydroxypyrazine (IBHP). Finally, the IBHP is methylated to IBMP. Whereas the other hypothesis proposed by Cheng et al. [23] believes that the MPs’ biosynthesis begins in a condensation of two amino acids, which forms a cyclic dipeptide, followed by several steps to convert into the HPs, and finally the HPs are methylated to MPs (hereinafter referred to as “Amino acid condensation pathway”). Take the 3-isopropyl-2-methoxypyrazine (IPMP) biosynthesis that had been studied in *Pseudomonas perolens*, for instance; the IPMP biosynthesis begins in the condensation of valine with glycine to form the 2,5-dicarbonyl-3-isopropyl-piperazine (DCPP), and then the DCPP was converted into the 3-isopropyl-2-hydroxypyrazine (IPHP). Finally, the IPHP was methylated to IPMP.

It is worth noting that the common point of the two hypotheses is both of them suggested that the final step of MPs’ biosynthesis is the HPs being methylated to MPs [24,25,26], which had been confirmed in grape. An O-methyltransferase (OMT), which is capable of methylating HPs to MPs, has been isolated and purified, and four OMT genes, including *VvOMT1*, *VvOMT2*, *VvOMT3*, and *VvOMT4*, had been identified in wine grapes [27,28,29]. The key difference between the two hypotheses is that the “Amino acid amidation pathway” hypothesized that the MPs’ biosynthesis begins with the amidation of a branched-chain amino acid [22], whereas the “Amino acid condensation pathway” hypothesized that the first step of MPs’ biosynthesis is a two-amino-acid condensation reaction to form a cyclic dipeptide [23]. Therefore, according to the “Amino acid condensation pathway”, the biosynthesis of IBMP begins in the condensation of leucine with glycine, which forms 2,5-dicarbonyl-3-isobutyl-piperazine (DCIP) (Figure 1). However, to date, except for the final step, the other information about MPs biosynthesis pathway still remains in the hypothesis stage. Once the results of relevant research are different from the general cognition [30,31,32], it is difficult to explain from the mechanism level.

Our previous study had explored the effect of the assumed precursor L-Leu and the possible key intermediate AMPA on IBMP biosynthesis in grape berry according to the “Amino acid amidation pathway” [22]. Here, we reported a further study on IBMP biosynthesis in wine grapes based on the “Amino acid condensation pathway” [23]. The DCIP that was proposed as the possible key intermediate of IBMP biosynthesis was incorporated into the Cabernet Sauvignon grapes at pre-veraison in *situ*. In order to avoid the influence of year difference on the results of the field experiments, the IBMP content in grape berries and the relative expression levels of VvOMTs in berry skins were monitored during the experimental period over two consecutive years (2020–2021). The effect of DCIP on IBMP accumulation in grape berry and the relative VvOMT genes expression pattern of berry skins was evaluated. The results would further broaden the understanding of IBMP biosynthesis pathway in wine grape.

## 2. Materials and Methods

### 2.1. Chemicals and Solutions

Solvents and reagents were of analytical reagent grade or higher. Sodium fluoride (NaF, 98.0% pure) used as sample extraction buffer was purchased from Guangfu Technology and Development Co., Ltd. (Tianjin, China). 3-Isobutyl-2-methoxypyrazine (IBMP, 99% pure) used as the standard was purchased from Sigma-Aldrich (St. Louis, MO, USA). 2,5-Dicarbonyl-3-isobutyl-piperazine (DCIP, 95% pure) used as soaking incorporation solution in *situ* was purchased from Aozeal Certified Standards (Alameda, CA, USA). Deuterium-labeled IBMP (dIBMP, 98% atom% ^2^H) used as internal standard was obtained from CDN Isotopes (Pointe-Claire, Que., Canada).

### 2.2. Incorporation Treatments on Clusters In Situ

#### 2.2.1. Vineyard Site

The experimental vineyard that belongs to the Shandong Academy of Agricultural Sciences located in Zhangqiu, Shandong, China (36°45′ N; 117°22′ E; elevation 51 m). Eight-year-old Cabernet Sauvignon vines, which were grafted on the 1103P rootstock, were used to carry out the experiments over two consecutive years (2020 and 2021). The vineyard soil type was typical loam soil, and it was managed under the traditional management. The vines’ training method was single cordon with a vertical shoot positioning trellis system, row spacing is 2.5 m, and plant spacing is 1.0 m, approximate south-north row direction.

#### 2.2.2. Experiment Treatments

In this study, three treatments were carried out: normal growth grapes (Mock); deionized water (Control) and DCIP solutions (5 mM) were fed into the grape clusters in *situ*, respectively at pre-veraison, i.e., 59 days after anthesis (DAA) in 2020 and 63 DAA in 2021. Three replicates were used for each treatment, with 60 clusters for each replicate. All treatments were arranged in randomized blocks in three rows.

A soaking incorporation method that was developed by Hayasaka et al. [33] and Chassy et al. [34] was utilized and modified slightly [35]. The key operation points were mainly as follows: Choosing the soft, light, and transparent plastic bag to wrap onto the grape clusters; using the transparent adhesive tape to entwine around the plastic bag gently so as to make it as close to the grape cluster as possible to minimize the dosage of the incorporation solution; adding the corresponding aqueous solution into the plastic bag for 48 h incorporation, then, after that, removing the plastic bags with scissors and letting the grape clusters dry naturally [35].

#### 2.2.3. Sample Collection and Pretreatment

In 2020, berry samples were collected at 59, 67, 72 (veraison), 88, 104, 121, 128, and 154 DAA (maturity). In 2021, berry samples were collected at 63, 69 (veraison), 76, 85, 93, 101, 118, and 153 DAA (maturity). All samples were collected randomly in triplicate at each sampling time point. Approximately 250 berries were collected per replicate and were immediately placed on ice in foam boxes that were returned to the laboratory within 2 h. Among them, 15 berries were randomly selected in each replicate for peeling berry skins using a scalpel and then frozen in liquid nitrogen immediately. The grape berry samples were stored at −40 °C, and the berry skin samples were stored at −80 °C prior to analysis.

### 2.3. IBMP Quantitation

According to the method that was developed by Koch et al. [36] and modified slightly [35], the detection and quantitation of IBMP in grape berries were achieved in all samples through the stable isotope internal standard method. Analysis of IBMP extracts was carried out with an HS–SPME coupled with an Agilent 7890B–5977A GC–MS (Agilent, Santa Clara, CA, USA) with a mass-selective detector. The SPME fiber (23-gauge, 2 cm divinylbenzene/Carboxen™/polydimethylsiloxane; DVB/CARB/PDMS) was exposed to the vial headspace to extract the IBMP. A CombiPAL RSI 85 autosampler (CTC Analytics, Zwingen, Switzerland) was used for incubation, extraction, and desorption. The main operation steps were as follows:

Approximately 30.0 g of frozen grape berries were weighed and recorded accurately. The number of grape berries in each replicate was also recorded, in order to express the IBMP content in grape berries both in nanograms per kilogram (fresh weight) and picograms per berry. The frozen berries were de-seeded with the scalpel when they had melted slightly; the flesh was ground into a fine powder in liquid nitrogen immediately and then was transferred to the centrifuge tube (50.0 mL). Then, 10 mL extraction buffer (2 mM NaF) containing 200 ng·L^−1^ internal standard solution was added into the centrifuge tube. The samples were homogenized and then placed in high-speed refrigerated centrifuge (Eppendorf 5804R, Hamburg, Germany) at 5000× *g* for 7 min at −4 °C. Next, 10.0 mL supernatant was placed into the headspace vial, which contained 3.0 g NaCl, and was put under room temperature to equilibrate for 1 h before HS-SPME-GC-MS.

The parameter setting detail of the CTC autosampler system and the GC-MS condition was same as the method we had established [35].

### 2.4. RNA Extractions, cDNA Synthesis, and Real-Time Polymerase Chain Reaction

Total RNA was extracted from berry skins using the General Plant Total RNA Extraction Kit (Bioteke, Beijing, China). cDNA synthesis was prepared by using a Reverse Transcription Kit (Tiangen, Beijing, China). The primers of *VvOMT1*, *VvOMT3*, and *VvOMT4* were according to Lei et al. [35]. The primers for *VvOMT2* were as follows: the forward primer was 5′-TCCGAGAAGATGGCTATGAG-3′, and the reverse primer was 5′-CTGCAAAGTTGGAATCTTTAA-3′.

The relative expression levels of VvOMTs were analyzed and calculated according to our established method [35]. The relative expression level of genes in different treatment samples were determined in 2020 and 2021.

### 2.5. Statistics

Univariate ANOVA was conducted using SPSS 20.0 software (IBM, Armonk, NY, USA). Significant variations were detected by Fisher’s least significant difference test (LSD) at *p* ≤ 0.05 and *p* ≤ 0.01. Line charts were drawn by using OriginPro 9.1 (Origin Lab Corporation, Northampton, MA, USA). The pathway graphs were prepared using Chemdraw 20.0 software (Cambridge Soft Corporation, Cambridge, MA, USA).

## 3. Results

### 3.1. IBMP Synthesis and Accumulation in Grape Berry

The IBMP concentration in grape berries (Figure 1) and the relative expression level of *VvOMT1*, *VvOMT2*, *VvOMT3*, and *VvOMT4* in berry skins (Figure 2) were monitored during the experimental period in 2020 and 2021.

Firstly, the IBMP accumulation pattern in grape berries after being incorporated with deionized water (control) was almost the same as the natural condition (Mock) during the whole experiment period no matter if it was calculated in nanograms per kilogram or picograms per berry in two consecutive years. These results showed that the soaking incorporation treatment did not significantly affect IBMP accumulation in grape berries.

The IBMP concentration in Control and Mock treatments showed a continuous downward trend during the experiment period over the 2 years (2020 and 2021). In 2020, the IBMP content peaked at 142.67 ng∙kg^−1^ (Figure 2a) and 101.45 pg∙berry^−1^ (Figure 2b) at 59 DAA. In 2021, the maximum concentration of IBMP occurred at 63 DAA at 167.04 ng∙kg^−1^ (Figure 2c) and 98.6 pg∙berry^−1^ (Figure 2d).

Secondly, the relative expression level of *VvOMT1*, *VvOMT2*, *VvOMT3*, and *VvOMT4* in grape berry skins between Control and Mock treatments had no significant difference at each sample time points during the experiment period over the two years further showed that the soaking incorporation treatment did not significantly affect IBMP biosynthesis in grape berry skins.

The relative expression level of *VvOMT1* peaked at 59 DAA in 2020 (Figure 3a) and 63 DAA in 2021 (Figure 3b), thereinto, it showed an increased trend at pre-veraison in 2020 (Figure 3a), and then declined quickly until harvest in two years (Figure 3). This trend was synchronized with IBMP accumulation in grapes.

The relative expression level of *VvOMT2* showed a fluctuating trend during the experiment period over the 2 years. In 2020, it was low at the early stage of the experiment, but spiked at 104 DAA and then declined rapidly at 121 DAA, increased at 128 DAA and then declined again until harvest (Figure 3c). In 2021, it was slightly increased at pre-veraison and then declined, but spiked at 101 DAA and then declined sharply at 118 DAA (Figure 3d). This trend did not show positive correlation with IBMP accumulation in grape (Figure 3c,d).

The relative expression level of *VvOMT3* showed the same trend during the experiment period over the 2 years. It peaked at 59 DAA in 2020 (Figure 3e) and 63 DAA in 2021 (Figure 3f), respectively, and fell sharply at veraison, and then continued to decline quickly until harvest, which was synchronous with IBMP accumulation in grape regardless of how the IBMP concentration was expressed.

The *VvOMT4* expression showed some difference between the 2 years. In 2020, it peaked at 59 DAA and then declined fluctuant until harvest (Figure 3g). In 2021, it showed a slight decline at pre-veraison, but increased after veraison, especially spiked at 93 DAA, and then declined sharply until harvest (Figure 3h). Therefore, the relative expression level of *VvOMT4* may have has less relation with IBMP accumulation in grape.

### 3.2. IBMP Accumulation in Grapes after DCIP Incorporation

The IBMP concentration in grape berries after DCIP incorporation continued to decline during the experimental period in the two consecutive years, regardless of whether the IBMP concentration was expressed in nanograms per kilogram or picograms per berry (Figure 4), and there was no obvious difference compared with the Control and Mock treatments at pre-veraison (59–72 DAA in 2020 and 63–69 DAA in 2021). After veraison (88–121 DAA in 2020, 76–118 DAA in 2021), the IBMP concentration after DCIP incorporation was increased by varying degrees over the Control and Mock treatments, but there was no significant difference among the treatments until at maturity, the IBMP content after DCIP incorporation increased significantly. Thereinto, in 2020, when the IBMP content was calculated in nanograms per kilogram (Figure 4a), the IBMP concentration of grape at 121 DAA after DCIP incorporation was increased by 106.53% compared with the Control and by 116.69% compared with the Mock treatment. When the IBMP content was calculated in picograms per berry (Figure 4b), it was increased by 115.33% over the Control and by 126.04% over the Mock treatment at 121 DAA. In 2021, compared with the Control treatment, it was increased by 105.75% and 145.10% after DCIP incorporation when the IBMP content was calculated in nanograms per kilogram (Figure 4c), and by 122.11% and 176.79% when the IBMP content was calculated in picograms per berry (Figure 4d) at 101 and 118 DAA, respectively. The IBMP content in grape berries was increased by 184.13% and 145.10% after DCIP incorporation when the IBMP content was calculated in nanograms per kilogram compared with the Mock (Figure 4c), and by 210.29% and 142.19% when the IBMP content was calculated in picograms per berry at 101 and 118 DAA, respectively.

It is noteworthy that regardless of whether the IBMP content was expressed in nanograms per kilogram or picograms per berry, DCIP incorporation significantly increased the IBMP concentration of grape over the Control and Mock treatments at maturity (121 DAA in 2020, 101 and 118 DAA in 2021) in two consecutive years. And at harvest, the IBMP concentration of grapes was all below the quantitation limit in the two consecutive years (154 DAA in 2020, 153 DAA in 2021).

### 3.3. The Relative Expression Level of VvOMTs in Berry Skins after DCIP Incorporation

The relative expression levels of VvOMTs were monitored in two consecutive years (2020 and 2021) to evaluate the impact of DCIP incorporation on IBMP biosynthesis.

The relative expression level of *VvOMT1* in grape berry skins after DCIP incorporation showed a different trend during the experiment period for the two different years. In 2020, the relative expression level of *VvOMT1* after DCIP incorporation was obviously increased at 121 and 128 DAA over the Control and Mock (Figure 5a). However, no significant difference was observed after DCIP treatment during the whole period in 2021 over the Control and Mock (Figure 5b).

For relative expression levels of *VvOMT2* and *VvOMT4* in grape berry skins after DCIP treatment, both showed no significant difference from the Control and Mock treatments during the experiment period in both years (Figure 5c,d).

The relative expression level of *VvOMT3* in grape berry skins after DCIP incorporation showed a different change during the experiment period over the 2 years, but overall, it showed an increased trend compared with the Control and Mock. In 2020, the expression of *VvOMT3* after DCIP incorporation was significantly increased at 67 DAA over the Control and Mock, but then significantly decreased down to the Control and Mock treatments level at 72 DAA. As the grape developed, the relative expression level of *VvOMT3* after DCIP incorporation was significantly higher than the Control and Mock treatments from 104 to 128 DAA, which reflects the significant elevation of IBMP concentration at 121 DAA (Figure 5e). In 2021, the relative expression level of *VvOMT3* after DCIP incorporation was significantly higher than the Control and Mock treatments from 69 to 85 DAA (Figure 5f), which was 16–49 days earlier than the significant increase in IBMP concentration. These results suggest that the relative expression level of *VvOMT3* in grape peel is more closely linked with IBMP biosynthesis.

## 4. Discussion

### 4.1. The Relationship between the DCIP and IBMP Accumulation

In 2020 and 2021, the IBMP content in all treatments showed a continuous downward trend during the experiment period, which means the experiment treatments time point was close to the peak IBMP accumulation period in the experimental grape berries. This phenomenon was similar to our previous study about Leu and AMPA effect on IBMP biosynthesis, which was carried out in the same vineyard [35]. What is more important, these results showed that the IBMP accumulated at pre-veraison and degraded after veraison until harvest, which was consistent with the finding of previous studies [26,36,37,38]. Moreover, the IBMP concentration of grapes in all treatments were all below the quantitation limit in the two consecutive years at harvest (154 DAA in 2020, 153 DAA in 2021). These results show that delaying the harvest time is really an effective way to reduce the IBMP content in wine grapes.

It is noteworthy in this study that, regardless of whether the IBMP content was expressed in nanograms per kilogram or picograms per berry, the incorporation of DCIP in *situ* significantly increased the IBMP concentration of grape at maturity (121 DAA in 2020, 101 and 118 DAA in 2021) for 2 consecutive years.

On the one hand, these results suggest that DCIP might participate in the biosynthesis of IBMP in grape berry. The reason why the IBMP concentration only increased at maturity compared with the Control and Mock treatments, it might be because there are series of steps between the DCIP and IBMP, and the DCIP converts into the corresponding key intermediate at a slow rate, so the IBMP concentration of grape berry would not increase immediately and showed significant increase until the later experiment period.

On the other hand, the IBMP concentration after DCIP treatment showed a significant increase at maturity (i.e., post-veraison) when it is generally believed that IBMP in grape berry is in a degradation stage [36,37,38,39] compared with the Control and Mock treatments at the same sampling time points. It further suggested that IBMP metabolism in grape berry might be a dynamic equilibrium process between the biosynthesis and degradation [35]. In other words, the degradation rate of IBMP in grape berry is faster than the biosynthesis rate after veraison under normal growth condition, so the IBMP content in grape berry showed a continued decline after veraison. Whereas, when the DCIP, which was considered to be the possible key intermediate of IBMP biosynthesis, was incorporated into the grape berry in *situ*, the IBMP biosynthesis process might be accelerated after veraison due to the exogenous DCIP incorporation. Thus, the IBMP content showed a significant increase compared with the Control and Mock treatments at the same time points.

In view of the results above, the effect of DCIP on IBMP biosynthesis in grapes deserves to be further explored.

### 4.2. Correlation of the Relative Expression Level of VvOMTs in Berry Skins with IBMP Biosynthesis

The research on the biosynthesis of MPs in grapes is still insufficient knowledge, but the final step involving the HPs being methylated into the MPs had been confirmed [24,25,26]; especially, the four VvOMTs involved in it had been identified in wine grapes [27,28,29]. At present, in all relative studies about the IBMP accumulation, the VvOMTs expression pattern is almost the only way to interpret the results from the mechanistic level [40,41].

In the grape berries, 72% of the IBMP is located in the berry skins [42], so the relative expression level of VvOMTs in grape berry skins were monitored during the experiment period in two consecutive years (2020 and 2021) in this study.

In 2021, the relative expression level of *VvOMT1* was synchronized with the IBMP accumulation in grape, but in 2020, the maximum level of IBMP accumulation was 8 days earlier than the maximum *VvOMT1* relative expression. After DCIP incorporation, the relative expression level of *VvOMT1* in berry skin was significantly up-regulated at 121 and 128 DAA in 2020, but showed no significant difference from the Control and Mock treatments in 2021.

The expression pattern of *VvOMT3* in berry skin was synchronized with IBMP accumulation in grape in two consecutive years, and the peak relative expression level of *VvOMT3* was almost synchronized with the maximum IBMP level. After DCIP incorporation, the relative expression level of *VvOMT3* in berry skins was remarkably increased at 67 DAA in 2020 and was significantly up-regulated from 69 to 85 DAA in 2021, which was 16–49 days earlier than the significant increase of IBMP accumulation in grape berries.

These results suggested that both the *VvOMT1* and *VvOMT3* play a significant role in regulating the IBMP accumulation; the IBMP content in grape berries was positively correlated with the up-regulated expression of *VvOMT1* [27,28], but in particular, the *VvOMT3* expression pattern was almost synchronized with the IBMP accumulation, which plays a major role in MPs biosynthesis in grape berries due to the fact that it had a high affinity for the HP precursors of MPs [29,41].

The relative expression levels of *VvOMT2* and *VvOMT4* did not show a positive correlation with IBMP accumulation in grape berries during the experimental period over the two years. This may be because *VvOMT2* favors the methylation of 3-isopropyl-2-hydroxypyrazine (IPHP) to 3-isopropyl-2-methoxypyrazine (IPMP) in grape berries [27]. The *VvOMT4* might have little relation to IBMP accumulation in grape, which might be why the expression level of *VvOMT4* always had been ignored in other studies about MPs’ accumulation.

Only a few studies have focused on the metabolic pathways of MPs’ biosynthesis. Interpreting the mechanism of relevant MPs study findings only from the VvOMTs expression pattern, which is involved in the final step of MPs’ biosynthesis pathway, has certain limitations. More research on MPs’ metabolic pathway is needed to further explore.

On the whole, the results drawn from this study will provide reference for further understanding and much deeper research on IBMP biosynthesis.

## 5. Conclusions

This study highlights the effect of DCIP on IBMP biosynthesis in Cabernet Sauvignon grapes in *situ*, and the findings reveal a significant increase in IBMP content of grape berries in response to DCIP incorporation compared with the Mock and Control at maturation in two consecutive years. The expression level of *VvOMT1* and particularly that of *VvOMT3* was obviously up-regulated in berry skin after DCIP incorporation and closely mirrored the IBMP accumulation pattern in two consecutive years. Therefore, we speculate that DCIP might participate in IBMP biosynthesis in the grapes and plays an important role in regulating IBMP accumulation. Given the importance of IBMP in determining the sensory characteristics of wine, further studies on the effects of DCIP are warranted.

## Figures and Tables

**Figure 1 foods-12-03258-f001:**
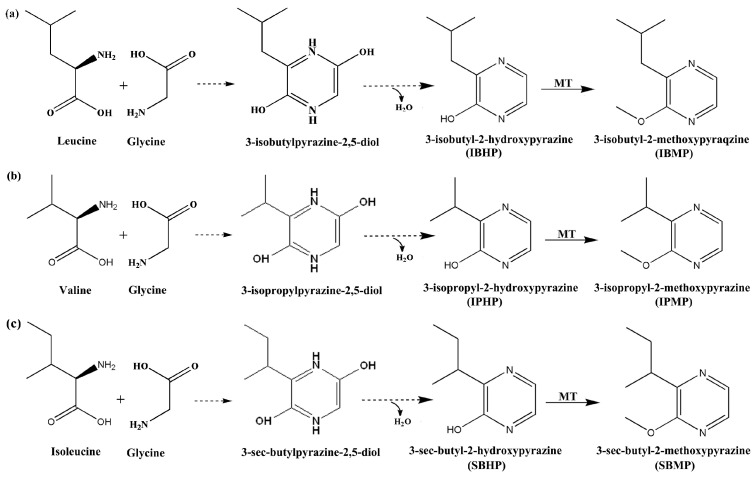
The proposed MPs biosynthesis through “Amino acid condensation pathway” by Cheng et al. [23]. (**a**) The condensation of leucine and glycine forms IBMP. (**b**) The condensation of valine and glycine forms IPMP. (**c**) The condensation of isoleucine and glycine forms SBMP. The dotted arrows indicate that the biosynthesis steps have not been demonstrated in living organisms. MT refers to methyltransferase.

**Figure 2 foods-12-03258-f002:**
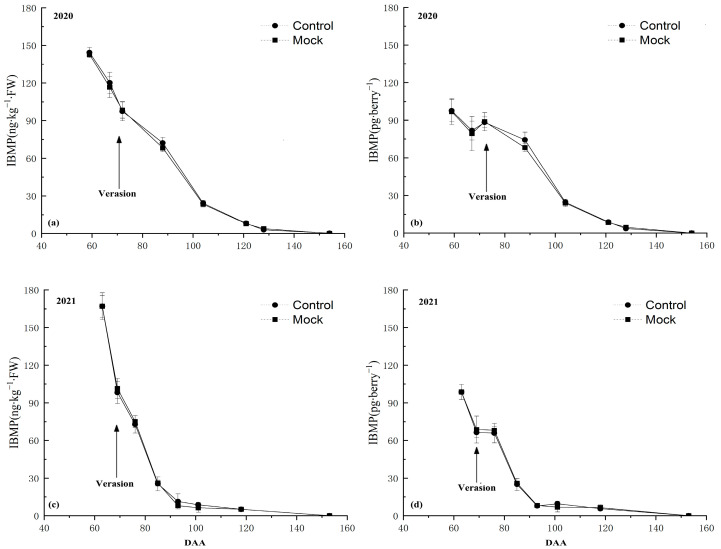
IBMP accumulation pattern in Cabernet Sauvignon grape berries over 2-year experimental period: (**a**) calculated in ng∙kg^−1^ and (**b**) pg∙berry^−1^ in the 2020 season. (**c**) IBMP content was calculated in ng∙kg^−1^ and (**d**) pg∙berry^−1^ in the 2021 season. BW, berry weight; DAA, days after anthesis. Data expressed as the average of three replicates ± standard error (SE).

**Figure 3 foods-12-03258-f003:**
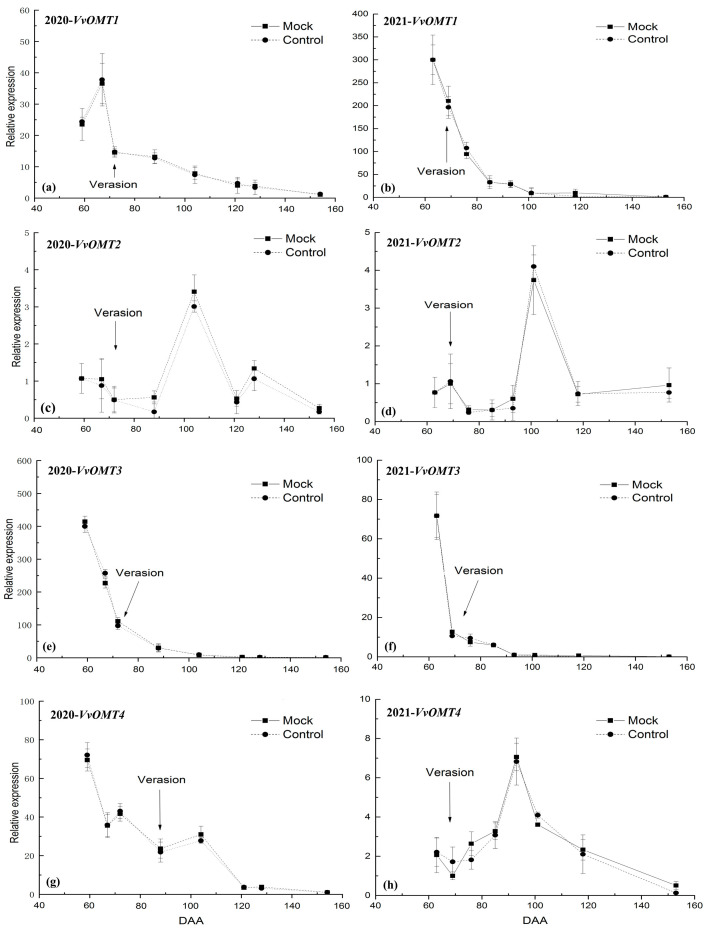
The relative expression level of VvOMTs in Cabernet Sauvignon berry skins during the experimental period: (**a**) *VvOMT1*, (**c**) *VvOMT2*, (**e**) *VvOMT3*, and (**g**) *VvOMT4* in the 2020 season, and (**b**) *VvOMT1*, (**d**) *VvOMT2*, (**f**) *VvOMT3*, and (**h**) *VvOMT4* in the 2021 season.

**Figure 4 foods-12-03258-f004:**
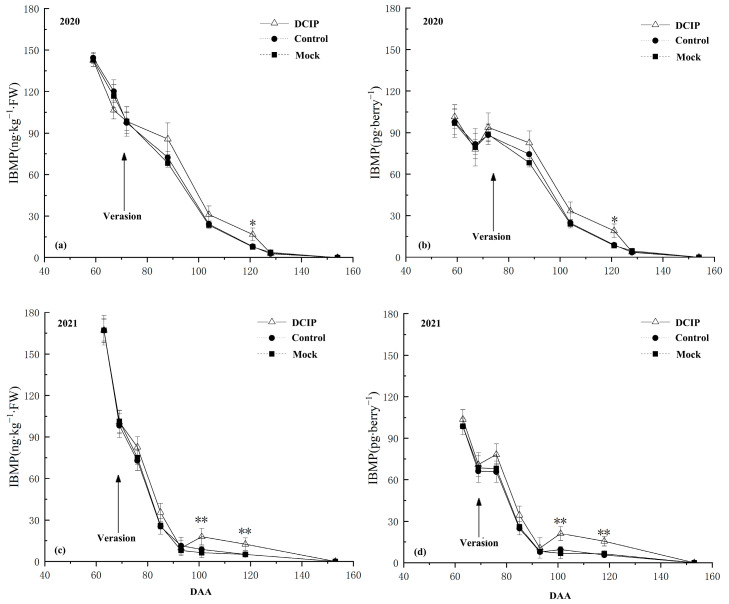
IBMP concentration after incorporation of 2,5-dicarbonyl-3-isobutyl-piperazine (DCIP) in Cabernet Sauvignon grapes during experimental period: (**a**) IBMP content calculated in ng∙kg^−1^ and (**b**) calculated in pg∙berry^−1^ in the 2020 season; (**c**) IBMP content calculated in ng∙kg^−1^ and (**d**) calculated in pg∙berry^−1^ in the 2021 season. The symbol * and ** indicate significant differences at *p* ≤ 0.05 and *p* ≤ 0.01 between control and DCIP treatment, respectively.

**Figure 5 foods-12-03258-f005:**
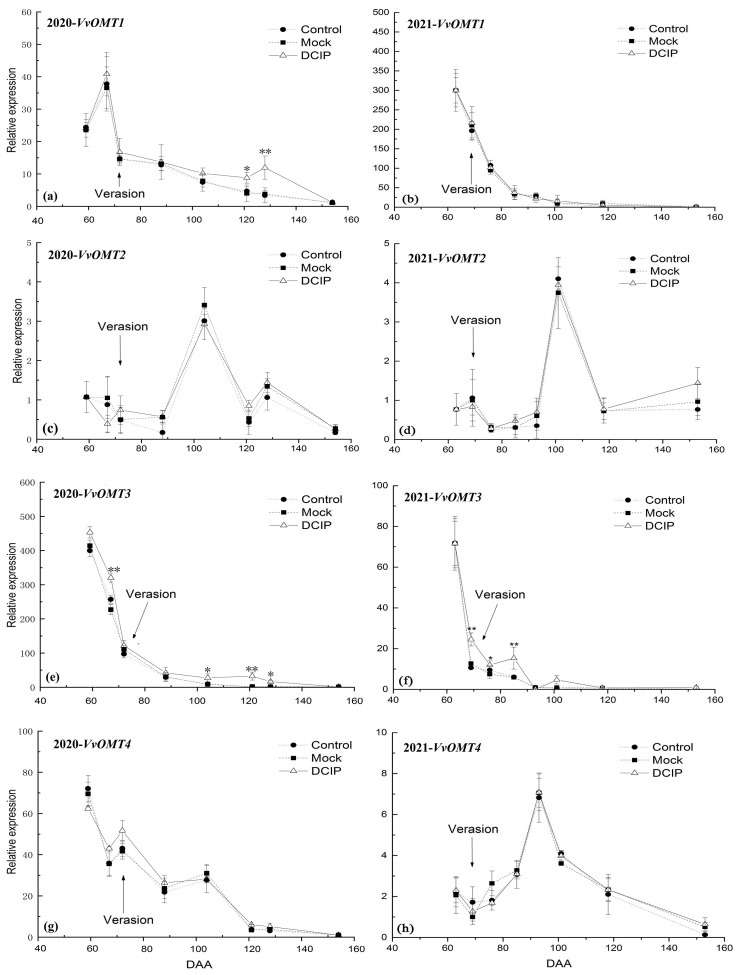
Effect of DCIP treatment on the relative expression levels of VvOMTs in berry skins during the experimental period: (**a**) *VvOMT1*, (**c**) *VvOMT2*, (**e**) *VvOMT3*, and (**g**) *VvOMT4* in the 2020 season, and (**b**) *VvOMT1*, (**d**) *VvOMT2*, (**f**) *VvOMT3*, and (**h**) *VvOMT4* in the 2021 season. The symbol * and ** indicate significant differences at *p* ≤ 0.05 and *p* ≤ 0.01 between control and DCIP treatment, respectively.

## Data Availability

The authors confirm that the data supporting the findings of this study are available within the article and the raw data that support the findings are available from the corresponding author upon reasonable request.

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
