# Peer review of "Effect of 2,5-Dicarbonyl-3-Isobutyl-Piperazine on 3-Isobutyl-2-Methoxypyrazine Biosynthesis in Wine Grape"

_foods, 2023, doi:10.3390/foods12173258_

Round 1
Reviewer 1 Report
This work aims at contributing to elucidate the mechanism of MPs biosynthesis in grapes, by proposing DCIP as key intermediate. The concept of the experiment is clear and the research has been performed by monitoring one wine grapes variety over two consecutive years.
Some remarks to improve the quality of the paper will be included herein:
- reasoning of choosing the 2 year timeframe needs to be mentioned in Section 1
- also in Section 1, several remarks on the analytical techniques choice and suitability for the experiment's scope (i.e. GC-MS, RT-PCR); same info to be included on the suitability and efficiency of the extraction procedures (i.e. SPME) for the as-harvested samples vs. available alternative techniques.
- Section 2, 2.1. and 2.3. to be improved by adding sufficient information on chemical analysis of IBMP; it is known that GC-MS together with related sample preparation sterp (i.e. extraction) is a very complex analysis, often influenced by the samples matrix, sample treatments, etc; normally the paper should provide a minimum set of information so that the experiment may be replicated by interested parties, with a reasonable further documentation work; in my opinion the indicated reference no. 30 provides some info, but not enough in this respect, as it refers to samples with different treatments; adding the employed quantities, volumes, timing, intermediate steps followed and other relevant info may improve the readers' understanding;
- not clear to me what are the results provided by the use of statistics software that are mentioned in section 2.5. (i.e. ANOVA)
- Section 3 - in my opinion there is an overlapped (doubled) information provided in Figure 5; in other words, Fig.5 includes all the information provided in Fig. 3 - to be checked and corrected as needed
Minor corrections needed throughout the paper
Author Response
Please see the attachment named "response_to_reviwer 1"

Reviewer 2 Report
In the paper, the authors investigate the influence of 2,5-dicarbonyl-3-isobutyl-piperazine on the expression of genes responsible for the synthesis of 3-isobutyl-2-methoxypyrazine in the skin of Cabernet Sauvignon cultivar. The results are very innovative and interesting. The work is relevant to future research on the synthesis of methoxypyrazine in grapes, about which there is still insufficient knowledge.
Author Response
Please see the attachment named "response_to_reviwer 2".

Reviewer 3 Report
In a very good manuscript of the article, the effect of 2, 5-dicarbonyl-3-isobutyl-piperazine (DCIP) on the biosynthesis of 3-isobutyl-2-methoxypyrazine (IBMP) in Cabernet Sauvignon grapes were investigated in situ.
The manuscript is written very systematically and without major errors. It contains a good review of relevant publications, the structure of the experimental work, the analytics, and includes 37 references. The results are adequately presented in five figures and are also satisfactorily processed statistically. A discussion of the results of the two-year experiment and the final conclusion of the research are adequate.
The results showed that IBMP levels in grapes treated with DCIP at the time of ripening were significantly higher than those of Control and Mock treatments. The expression level of VvOMT1 and especially VvOMT3 was significantly up-regulated in the berry skin after DCIP incorporation and accurately reflected the IBMP accumulation pattern in two consecutive years. Therefore, authors suggest that DCIP may be involved in IBMP biosynthesis in grapes and plays an important role in regulating IBMP accumulation. Given the importance of IBMP in the sensory characteristics of wine, further studies on the effects of DCIP are warranted.

Author Response
Please see the attachment named "response_to_reviwer 3".
